Transformer-based ensemble model for dialectal Arabic sentiment classification

Mansour Omar
Aboelela Eman eman.aboelela@intellaworld.com
Talaat Remon
Bustami Mahmoud
Intella , Cairo , Egypt
Benítez-Andrades José Alberto
Electronic publication date: 2025 Mar 24
Publication date: 2025
Volume: 11
Electronic Location ID: e2644
Received 2024 Jun 12; Accepted 2024 Dec 12
Copyright: © 2025 Mansour et al.
Copyright year: 2025
Copyright holder: Mansour et al.
License: This is an open access article distributed under the terms of the Creative Commons Attribution License, which permits unrestricted use, distribution, reproduction and adaptation in any medium and for any purpose provided that it is properly attributed. For attribution, the original author(s), title, publication source (PeerJ Computer Science) and either DOI or URL of the article must be cited.
License URL: https://creativecommons.org/licenses/by/4.0/

Keywords: Ensemble learning, Deep learning, Machine learning, CAMeLBERT, XLM-RoBERTa, MARBERT, FastText, Arabert, Aravec, Arabic sentiment classification

Funding: Intella This study was financially supported by Intella, which supplied the essential resources for carrying out the experiments and achieving the results discussed in this article. The funder had a role in study design, data collection and analysis, decision to publish, and preparation of the manuscript.

==============================
Social media platforms such as X, Facebook, and Instagram have become essential avenues for individuals to articulate their opinions, especially during global emergencies. These platforms offer valuable insights that necessitate analysis for informed decision-making and a deeper understanding of societal trends. Sentiment analysis is crucial for assessing public sentiment toward specific issues; however, applying it to dialectal Arabic presents considerable challenges in natural language processing. The complexity arises from the language’s intricate semantic and morphological structures, along with the existence of multiple dialects. This form of analysis, also referred to as sentiment classification, opinion mining, emotion mining, and review mining, is the focus of this study, which analyzes tweets from three benchmark datasets: the Arabic Sentiment Tweets Dataset (ASTD), the A Twitter-based Benchmark Arabic Sentiment Analysis Dataset (ASAD), and the Tweets Emoji Arabic Dataset (TEAD). The research involves experimentation with a variety of comparative models, including machine learning, deep learning, transformer-based models, and a transformer-based ensemble model. Feature extraction for both machine learning and deep learning approaches is performed using techniques such as AraVec, FastText, AraBERT, and Term Frequency-Inverse Document Frequency (TF-IDF). The study compares machine learning models such as support vector machine (SVM), naïve Bayes (NB), decision tree (DT), and extreme gradient boosting (XGBoost) with deep learning models such as convolutional neural networks (CNN) and bidirectional long short-term memory (BLSTM) networks. Additionally, it explores transformer-based models such as CAMeLBERT, XLM-RoBERTa, and MARBERT, along with their ensemble configurations. The findings demonstrate that the proposed transformer-based ensemble model achieved superior performance, with average accuracy, recall, precision, and F1-score of 90.4%, 88%, 87.3%, and 87.7%, respectively.

Introduction

The utilization of natural language processing (NLP) has broadened across a multitude of fields, including entertainment, politics, customer service, finance, and social media, with particular focus on sentiment classification (Matrane, Benabbou & Sael, 2023). This process entails discerning positive, negative, or neutral sentiments within unstructured text, encompassing various Arabic dialects.

Numerous studies have focused on sentiment classification in languages such as English, Spanish, Italian, and French; however, research concerning the Arabic language remains relatively sparse, despite its usage across 22 countries by over 330 million people (Abdelminaam et al., 2021). The Arabic language encompasses various forms, including Classical Arabic, Modern Standard Arabic (MSA), and dialectal Arabic, each exhibiting unique characteristics. Developing a generalized model that accommodates the diverse forms and dialects of Arabic is particularly challenging due to the lack of standardized orthographies in dialectal Arabic, as well as the syntactic, morphological, lexical, and phonological disparities between dialectal Arabic and MSA (Alharbi, 2019). Notably, a single word may convey different syntactic meanings across dialects. For example, the word “want” is represented as “تريد” or “toreed” in Jordanian Arabic, “بدك” or “biddak” in Saudi Arabic, and “عايز” or “aa-yez” in Egyptian Arabic.

The sentiment classification process involves two primary phases: feature engineering, which transforms text into a numerical representation, and the classification of dialectal Arabic review sentiments, which assesses the review’s polarity as either positive, negative, or neutral. This process comprises three main components: opinion holder, opinion object, and opinion orientation (Aboelela, Gad & Ismail, 2021a). The opinion holder is defined as anyone who expresses their views, while the opinion object refers to the feature that the opinion holder can discuss. Opinion orientation represents the sentiment polarity of the expressed opinion, categorized as positive, negative, or neutral. Dialectal Arabic sentiment classification has been analyzed at three main levels: document or review level, sentence level, and aspect or feature-based level, thereby providing insights into the sentiment polarity of reviews (Alharbi & Sharma, 2024). At the document level, the dialectal Arabic review focuses on an opinion concerning a single entity, which is classified as either positive, negative, or neutral. For example, the following Qatari review is classified as positive: “مكان جيد و رخيص و لكن الزحمة وايييييد ، و كثرة الهنود احسة كانه بومباي و لكن جيد و الادوات الكهربائية رخيصة فيه …يحتاج الى توسعة المواقف للسيارات”, which translates in English to “It’s a nice and cheap place, but it’s very crowded, and there are a lot of Indians—feels like Bombay. Overall, it’s good, and the electrical tools are cheap, but it needs an expansion of the parking spaces”. At the sentence level, sentiment classification evaluates review sentences independently, treating each sentence as having a distinct opinion. Each sentence is categorized as either positive, negative, or neutral. For instance, the following Kuwaiti review is classified as negative: “المسافة ما بين صفتات السيارات ومكان الدخول للجزيرة بعيد جدا يريت يكون فيه خدمة لتقريب المسافة لوجود اغراض مع المنتزهين …وشكراً”, which translates to “The distance between the car rows and the entrance to the island is very far. It would be great if there were a service to shorten the distance, especially since visitors have items with them. Thank you”. Aspect-level sentiment classification is crucial for precise decision-making, as it examines the aspects highlighted in the review to determine sentiment polarity (positive or negative). This approach is essential for identifying the positive and negative attributes of a service or product (Aboelela, Gad & Ismail, 2021b). For instance, the Egyptian review “البلوتوث بتاعه لاب سعره ممتاز بالنسبة للسوق مشكلتة بس فى” translates to “The laptop has an excellent price compared to the market, but the issue is with its Bluetooth,” and is classified as positive concerning the aspect “price” or “السعر” and negative with regard to the aspect “bluetooth” or “البلوتوث”.

The sentiment classification process encompasses three primary methodologies: machine learning methods, lexicon-based methods, and hybrid methods. Machine learning methods can be categorized into three types of learning: supervised learning, unsupervised learning, and semi-supervised learning. The supervised learning approach relies on labeled data, while the unsupervised learning method utilizes unlabeled data. Semi-supervised learning combines elements of both techniques. In contrast, lexicon-based methods employ a corpus of sentiment words to ascertain the polarity of reviews. Hybrid methods aim to merge the accuracy of machine learning techniques with the efficiency of lexicon-based approaches (Nasayreh et al., 2024). This article concentrates on sentiment classification at the document or review level using the supervised machine learning technique.

Sentiment classification has been addressed through a variety of techniques, including machine learning, deep learning, transformer learning, and ensemble learning. In machine learning-based sentiment classification, textual data undergoes pre-processing and feature engineering before being analyzed by traditional classifiers such as naïve Bayes (NB) and support vector machines (SVM). Conversely, deep learning-based sentiment classification employs pre-trained embedding models such as GloVe and Word2Vec to cleanse and encode the text, which is then input into deep learning architectures such as long short-term memory (LSTM), recurrent neural networks (RNNs), and gated recurrent units (GRUs) for classification purposes (Tan, Lee & Lim, 2023). Transformer models have emerged as a superior alternative to deep learning models, utilizing an encoder stack and a decoder stack with attention mechanisms to achieve enhanced performance (El Karfi & El Fkihi, 2022). The encoder stack is responsible for extracting features from the input sentence, while the decoder stack processes these features. In ensemble learning-based sentiment classification, multiple models are integrated using techniques such as Bootstrap aggregating (bagging), boosting, and voting to enhance performance (Akano & James, 2022). Bagging involves training a group of weak classifiers in parallel and merging their outputs through deterministic averaging, while boosting trains classifiers sequentially and combines them using a strategic approach. Voting aggregates the predictions of the trained models to generate a final outcome.

This article introduces a transformer-based ensemble learning model specifically designed for sentiment classification of dialectal Arabic. Both ensemble learning methods and transformers have demonstrated significant improvements in the performance of dialectal Arabic text classification. The study comprises three distinct types of experiments. The first type utilizes optimized machine learning algorithms, including SVM, NB, decision tree (DT), and extreme gradient boosting (XGBoost). The second type employs optimized deep learning algorithms such as convolutional neural networks (CNN) and bidirectional long short-term memory (BiLSTM). The third type involves an ensemble learning model that integrates three transformer-based architectures: Contextualized Arabic Language Model BERT (CAMelBERT), Cross-lingual Model-RoBERTa (XLM-RoBERTa), and Multilingual Arabic BERT (MARBERT). The feature extraction process is critical for both machine learning and deep learning models, as it directly influences their performance. To this end, various feature extraction methods are evaluated, including AraVec, FastText, Term Frequency-Inverse Document Frequency (TF-IDF), and Arabic BERT (AraBERT), which account for both textual and semantic features of words.

In summary, this article provides a thorough analysis of the advancements in sentiment classification within dialectal Arabic, accomplished through extensive experimentation with machine learning, deep learning, and transformer-based ensemble models. The study assesses numerous machine learning algorithms, including SVM, NB, DT, and XGBoost. For the deep learning segment, both CNN and BLSTM models are utilized. Following this, the research proposes an ensemble approach that leverages the strengths of three transformer-based models: CAMeLBERT, XLM-RoBERTa, and MARBERT. Furthermore, the study investigates various feature engineering techniques, including the development of an Arabic emoji mapping lexicon to be employed during the pre-processing stage, which significantly enhances model performance. The hyper-parameters for the proposed ensemble model and deep learning architectures are optimized using the Optuna library in conjunction with the Adam optimizer, while the hyper-parameters for the machine learning models are fine-tuned through grid search.

The organization of this article is outlined as follows: “Related Work” examines relevant studies on dialectal Arabic sentiment classification. “Methodology” details the methodology utilized in this research. The findings and analysis are presented in “Results and Analysis”. A discussion of the results is provided in “Results Discussion”. Finally, “Conclusion And Future Work” concludes the article and highlights directions for future work.

Related work

The research conducted in the field of sentiment classification for dialectal Arabic can be categorized into four main techniques:

Rule-based techniques

In Nasser & Sever (2020), a sentiment analyzer specifically designed for Arabic is introduced, focusing on sentence-level concepts. The authors utilized a semantic parser along with a rule-based concept extraction method that leverages Arabic grammatical rules and morphological analysis. They implemented four feature extraction techniques: concept-based features (CBF), Bag of Words features (BoW), lexicon-based features (LEX), and Word2Vector features (W2V). For the sentiment classification task, the following approaches are employed: (1) the Ar-SenticNet sentiment lexicon, (2) a variety of machine learning algorithms including SVM, hidden Markov model (HMM), naïve Bayes (NB), and linear regression (LR), and (3) combinations of classification algorithms, such as SVM-HMM, SVM-LR, and SVM-NB. The evaluation of the proposed sentiment analyzer is conducted using the GDELT Large-scale Arabic Sentiment Corpus (GLASC), which comprises 620,082 Arabic news articles.

In Diwali et al. (2022), a hybrid framework is introduced for sentiment analysis that integrates Arabic dependency-based linguistic patterns with deep neural network models. The results obtained from this framework exceeded those of SVM, LR, and various deep learning models such as long short-term memory (LSTM), CNN, and BiLSTM. The experiments are carried out on three benchmark datasets: the Arabic Sentiment Tweets Dataset (ASTD), the Arabic Jordanian General Tweets (AJGT) dataset, and the ArTwitter dataset. Lexicons based on NileULEX, Ar-SenticNet, and the Large-scale Arabic Book Review (LABR) dataset are utilized in the sentiment classification process. The authors proposed that if the linguistic rules were unable to determine the sentiment score, the deep learning models would then be employed.

The authors in El-Beltagy et al. (2018) and Shoukry & Rafea (2015) employed the NileULex sentiment lexicon to extract lexical features from Arabic text, which are subsequently used as inputs for an SVM classifier. The experiments are conducted on datasets featuring Modern Standard Arabic (MSA). Likewise, in Aldayel & Azmi (2016), the same methodology is applied to reviews written in dialectal Arabic.

Baseline machine learning techniques

A framework for “context-sensitive sentiment analysis” specifically designed for the Arabic language is presented in Fouda et al. (2024). This framework addresses the challenges associated with linguistic diversity and dialectal variations present in the language. It includes an extensive pre-processing phase aimed at standardizing the dataset by removing diacritics, translating emojis into textual descriptions, and normalizing the text. The study employs logistic regression and naive Bayes models, which are tailored to capture the unique linguistic characteristics of Arabic. The results reveal that positive sentiments dominate the analyzed reviews, comprising 59.9%, while neutral sentiments account for 35.4% and negative sentiments make up 4.7%. The authors underscore that their framework significantly enhances the understanding of customer sentiments in Arabic-speaking markets and sets a new benchmark for sentiment analysis in natural language processing.

In Alowaidi, Saleh & Abulnaja (2017), the authors present a semantic Arabic Twitter Sentiment Analysis (ATSA) model that employs supervised machine learning techniques in conjunction with semantic analysis. They explore the influence of employing both lexical and semantic feature representations on the sentiment analysis process. The authors implement the BoW method for lexical representation and utilize Arabic WordNet for semantic representation. The evaluation is carried out using a dataset consisting of 413 positive and 413 negative Arabic tweets, with SVM and NB classifiers employed for analysis. The experimental results demonstrate that the incorporation of semantic feature representation significantly outperforms the baseline BoW model.

In Al-Obaidi & Samawi (2016), the authors proposed a sentiment analysis model named SAMCAL specifically designed for a colloquial variety of the Arabic language. The SAMCAL model consists of three primary steps: (1) pre-processing, (2) review representation and feature extraction, and (3) classification. During the pre-processing step, stop words are removed, normalization is applied, and dialectal light stemming is performed. For the feature extraction phase, the researchers utilize the BoW and N-gram (3-gram phrases) techniques. In the classification step, they employ naïve Bayes (NB), SVM, and maximum entropy methods. The experimental results indicate that the Maximum Entropy method outperformed both NB and SVM classifiers. The evaluation of the models is conducted on a dataset comprising 28,576 reviews from Jordan and Saudi Arabia.

In Louati et al. (2023), a technique known as SVM-SAA-SC (Support Vector Machine Sentiment Analysis for Arabic Course Reviews) is introduced to evaluate student feedback in higher education, specifically focusing on Arabic text. This method employs a comprehensive pre-processing strategy that encompasses data collection, removal of irrelevant information, tokenization, stop word elimination, and stemming. Sentiment classification is then performed using the SVM algorithm. The model is trained on a dataset of 384 student reviews from Prince Sattam bin Abdulaziz University (PSAU). The SVM-SAA-SC model achieved an accuracy rate of 84.7%, with approximately 69.62% of the reviews classified as positive. To evaluate its effectiveness, the performance of the SVM-SAA-SC model is compared with that of the CAMeLBERT Dialectal Arabic model, which classified 70.48% of the reviews as positive. Although CAMeLBERT attained a slightly higher positive classification rate, the SVM-SAA-SC model demonstrated strong performance and effectiveness in sentiment analysis, highlighting its potential as a valuable tool for enhancing educational insights in the realm of Arabic higher education.

In Abdallah & Abo-Suaileek (2019), a framework for sentiment analysis specifically tailored for slang Arabic text is introduced. The authors developed a novel political dataset from Jordan, comprising 2,000 tweets categorized as either positive or negative, written in both slang and standard Arabic. To improve the performance of tweet classification, they implemented four distinct feature types: writing style, lexicon, emotional, and grammatical features. A variety of machine learning algorithms are employed during the classification process, including RF, Dagging, multi-class classifier (MC), regression (CVR), NB, simple logistic (SL), and MultiBoost.

In a study conducted by Al Sari et al. (2022), sentiment analysis is utilized to evaluate the opinions of passengers and viewers regarding their initial experiences in Saudi Arabia during a cruise. An Arabic dataset is compiled from three platforms: Instagram, Snapchat, and X. Initially, the dataset contained 10,922 instances, which are subsequently reduced to 1,200 through a thorough cleaning process. Feature extraction is carried out using the N-Gram method, and the opinions are classified using five different machine learning algorithms: NB, multi-layer perceptron (MLP), voting ensemble algorithm, SVM, and RF. The findings indicated that both NB and MLP outperformed the other classifiers.

The authors of Nasayreh et al. (2024) conducted an analysis of an Arabic X dataset to evaluate public opinion regarding the use of ChatGPT. This dataset included 1,473 positive tweets and 774 negative tweets. To tackle the problem of imbalanced data, they utilized the Synthetic Minority Oversampling Technique (SMOTE) (Chawla et al., 2002). The tweets underwent a cleaning process, and the TF-IDF technique is employed to identify the most significant features within the reviews. For the classification of these reviews, several machine learning models are implemented, including RF, LR, NB, and SVM. The results indicated that the SVM algorithm outperformed the other models. Additionally, hyper-parameter optimization is performed using Grid Search, Random Search, and Bayesian Optimization techniques.

Deep learning techniques

In Zaied & Soliman (2024), an “Arabic Sentiment Analysis Framework” is presented, which integrates deep learning and conventional machine learning techniques to effectively evaluate sentiments within Arabic texts. This framework encompasses various models, including LR, SVM, multinomial naïve Bayes (MNB), RF, DT, as well as sophisticated deep learning methods such as LSTM, CNN, and a hybrid CNN-LSTM model. For feature extraction, the study employed TF-IDF and Word2Vec, facilitating the classification of sentiments into three distinct categories: positive, neutral, and negative. The results indicate that both the LR and SVM classifiers achieved the highest accuracy of 87%, outperforming models such as LSTM, which attained 86.41%, and CNN, which reached 85.26%. The analysis is conducted using the Arabic Companies Reviews dataset, providing a robust basis for assessing the proposed method’s effectiveness compared to existing strategies in the domain.

In Aoumeur, Li & Alshari (2023), the authors developed a new classical Arabic dataset known as CASAD by utilizing various books, with human experts responsible for labeling the sentences. To effectively weigh and semantically represent the features for training, they implemented techniques such as TF-IDF, count vectors, and Word2Vec for feature extraction. The study employed six machine learning algorithms, including SVM, NB, LR, KNN, Classification and Regression Trees (CART), and Latent Dirichlet Allocation (LDA), to classify the sentences of the stories. The classification approach focused on detecting both binary classes (positive and negative) and multi-class scenarios (positive, negative, and neutral). The findings revealed that the performance of the classifiers for binary class detection was superior to that for multi-class categorization.

The authors in Elhassan et al. (2023) proposed a deep learning framework to illustrate the effectiveness of utilizing deep neural networks and word embedding techniques in sentiment analysis. They compared two word embedding methods: Skip-Gram (SG) from FastText and Continuous Bag of Words (CBOW) from Word2Vec, to identify the most effective feature representations. For the task of review classification, three deep neural network architectures are employed: CNNs, LSTM, and a hybrid CNN-LSTM model. The performance of these models is assessed using two benchmark datasets: the Hotel Arabic Reviews Dataset (HARD) and Large-Scale Arabic Book Reviews (LARB), evaluating various factors such as balanced vs unbalanced datasets and binary vs multi-class classification. The results indicated that the highest accuracy is obtained using FastText on the HARD two-unbalanced dataset, while the LARB three-unbalanced dataset recorded the lowest accuracy with both FastText and Word2Vec.

In Elsamadony, Keshk & Abdelatey (2023), a deep learning architecture based on bidirectional long short-term memory (BLSTM) is proposed, utilizing the Large-Scale Arabic Book Reviews (LARB) dataset, which contains 63,000 comments. The study compared three feature extraction methods: one-hot encoding, TF-IDF, and Word2Vec, specifically using the Skip-Gram and Continuous Bag of Words techniques. Among these methods, Word2Vec demonstrated superior performance. For classification, seven machine learning techniques are employed, including random forest, logistic regression, support vector machine, k-nearest neighbors, XGBoost, and decision tree, with XGBoost yielding the best results. Additionally, three deep learning models—BLSTM, CNN, and LSTM are evaluated. The findings revealed that the combination of BLSTM with Word2Vec outperformed XGBoost when paired with Word2Vec.

In Alhumoud et al. (2023), the authors utilized the largest Arabic human-annotated corpus related to the COVID-19 vaccine, known as ASAVACT, which comprises 32,476 tweets manually annotated by seven language experts. Their objective was to classify public opinions from the Gulf region regarding the COVID-19 vaccine into three categories: anti-vaccine, pro-vaccine, and neutral. To accomplish this, they employed two deep learning models: the stacked gated recurrent unit (SGRU) and the stacked bidirectional gated recurrent unit (SBi-GRU). The performance of these deep learning classifiers is compared against several machine learning algorithms, including NB, SVM, RF, LR, and DT. The results indicated that the deep learning models outperformed the traditional machine learning techniques. Furthermore, the accuracy is significantly enhanced by employing an ensemble architecture that combined SGRU, SBi-GRU, and AraBERT, resulting in an improvement of at least 7%.

In Heikal, Torki & El-Makky (2018), the authors employed a combination of deep neural networks, specifically LSTM and CNN, to assess the polarity of Arabic tweets. They evaluated a total of 3,315 tweets from the ASTD dataset, applying various hyper-parameter tuning techniques to identify the optimal models for both CNN and LSTM. For feature extraction, the authors utilized AraVec and Word2Vec.

Transformer-based techniques

An ensemble learning model is introduced in Mohamed et al. (2022) for Arabic sentiment classification, integrating both the multilingual XLM-T and the monolingual MARBERT models. To tackle the challenge of imbalanced data, the authors implemented focal loss and label smoothing techniques. They fine-tuned the model using three datasets: the Twitter-based Benchmark Arabic Sentiment Analysis Dataset (ASAD), SemEval-2017, and ArSarcasm-v2. The results indicated that the ensemble model surpassed all other models in performance.

In Alosaimi et al. (2024), the authors introduced a method called “AraBERT-LSTM” designed to enhance sentiment analysis for Arabic texts. This innovative approach integrates the strengths of the AraBERT transformer model with LSTM networks, utilizing AraBERT as the foundational layer to provide rich contextual embeddings that are specifically tailored for Arabic language processing. The LSTM layer further ensures the effective capture of long-term dependencies within the dataset. The research employs specific pre-processing techniques such as tokenization and normalization, along with three different word embedding strategies: Continuous Bag-of-Words (CBOW), Skip-Gram, and AraBERT embeddings. The performance of the AraBERT-LSTM model is rigorously compared against several machine learning and deep learning algorithms, including decision trees, random forest, logistic regression, k-nearest neighbors, naïve Bayes, GRU, and LSTM. Results demonstrated that the AraBERT-LSTM model achieved impressive accuracy exceeding 97%, outpacing all other evaluated models, with a noteworthy accuracy of 90.40% on the SS2030 dataset, thereby underscoring its effectiveness in the domain of Arabic sentiment analysis.

In El Karfi & El Fkihi (2022), the authors conducted experiments using CAMeLBERT, AraBERTv0.2, and a majority voting model to perform sentiment analysis. They pre-processed the data by removing Arabic stop words, normalizing the text, and eliminating all emoticons and emojis. The models are fine-tuned across four distinct datasets: the X dataset, which included 2,000 tweets from Modern Standard Arabic (MSA) and Jordanian sources; the Arabic Gold-Standard X dataset containing 6,512 tweets; the Arabic Sentiment Tweets Dataset (ASTD), comprising 10,006 tweets in both MSA and Arabic dialects; and a collection of 1,299 tweets from Modern Standard Arabic book reviews. The ensemble model yielded promising results, showcasing its effectiveness in the sentiment analysis task.

In Gaanoun & Benelallam (2021), the authors proposed an ensemble learning model that integrates three distinct models: Gaussian naive Bayes, MARBERT, and BiLSTM utilizing Mazajak embeddings. They fine-tuned these models using the ArSarcasm-v2 dataset and implemented a weighted ensemble approach to enhance the accuracy of their predictions.

In Hassan (2024), Arabic tweets regarding the Chinese president’s visit to Saudi Arabia are analyzed. The authors filtered out any tweets containing specific phrases such as: “زيارة الرئيس الصيني”, “الصين, الرئيس الصيني”, “القمة السعودية الصينية”, “القمة الخليجية الصينية”, “القمة العربية الصينية”, and “السعودية” which translates in English “China”, “Visit of the Chinese President”, “Chinese President”, “Saudi-Chinese Summit”, “Gulf-China Summit”, “Arab-Chinese Summit”, and “Saudi Arabia” respectively. The final dataset comprised 10,361 tweets posted between December 1, 2022, and December 14, 2022. Various pre-processing steps are carried out, including the removal of spam and duplicate tweets, as well as the deletion of emojis, URLs, emoticons, mentions, punctuation, hashtags, and non-Arabic text, in addition to the normalization of elongated letters. For the classification of the tweets, a pre-trained CAMeLBERT-Dialectal Arabic Sentiment Analysis (CAMeLBERT-DA SA) model is utilized.

The authors in Wadhawan (2021) utilized two transformer-based models, AraELECTRA and AraBERT, to detect sarcasm and sentiment. For evaluation, they employed the ArSarcasm-v2 dataset, which underwent several pre-processing steps, including the removal of HTML line breaks and markup, as well as the replacement of all URLs. The findings indicated that the AraBERT model outperformed AraELECTRA in both sarcasm and sentiment identification tasks.

This study distinguishes itself from previous research in several key ways: it emphasizes the lexical and semantic representation of features by employing various feature engineering techniques, including word embeddings such as AraVec and FastText, transformer architectures such as AraBERT, and the TF-IDF vectorizer.

it implements hyper-parameter optimization techniques for deep learning, machine learning models, transformer-based models, and their ensemble counterparts.

the study develops an Arabic emoji mapping lexicon consisting of 463 positive and negative emojis, which is utilized during the tweets pre-processing phase.

it introduces an optimized transformer-based ensemble model that, for the first time in dialectal Arabic sentiment classification studies, integrates three transformer-based models: CAMeLBERT, XLM-RoBERTa, and MARBERT.

Methodology

In this section, we outline our proposed approach for sentiment classification of dialectal Arabic. The process begins with the pre-processing of tweets, after which tasks are initiated based on the type of model employed. For deep learning or machine learning models, features are extracted using TF-IDF for machine learning models, while AraVec, FastText, and AraBERT are used for deep learning models. These extracted features are then fed into the respective models for sentiment classification. In the case of transformer-based models, the pre-processed tweets are directly inputted into the proposed transformer-based ensemble model for sentiment identification. Following this, we evaluate the performance of the models and apply hyper-parameter tuning techniques to enhance their performance. The models are subsequently fine-tuned using optimal parameters and re-evaluated (refer to Fig. 1). The upcoming sections provide a comprehensive analysis of the features of each module within the proposed approach.

Figure 1 The workflow of the proposed methodology.

Tweets gathering

Before initiating data collection for our analysis, we established specific criteria to guide our focus. We aimed to gather data from a social media platform that is written in Arabic, included emojis, and represented multiple dialects and aspects. Given these parameters, it was impractical to collect, clean, and prepare a substantial volume of raw data for sentiment annotation within a reasonable timeframe. As a result, we concentrated on three distinct public Arabic social media datasets, all composed of tweets from the X platform. This collection process yielded datasets rich in Arabic dialects from various regions, including the Levant (comprising Syria, Lebanon, Jordan, and Palestine), the Gulf (including Saudi Arabia, Kuwait, Qatar, Bahrain, the United Arab Emirates, and Oman), as well as Egypt, Sudan, and the Maghreb (covering Libya, Algeria, Tunisia, and Morocco). Collectively, these three datasets encompass seven dialects—TEAD (four dialects), ASAD (four dialects), and ASTD (one dialect)—spoken across 18 of the 22 Arab countries. The forthcoming subsections, along with Table 1, will provide a detailed overview of the relevant information regarding these three datasets. Arabic Sentiment Tweets Dataset (ASTD): is an Arabic social sentiment analysis resource derived from X and can be accessed at https://github.com/mahmoudnabil/ASTD/tree/master/data. It comprises a total of 10,006 tweets classified according to their sentiment as positive, negative, or neutral. Specifically, the dataset includes 799 positive tweets, 1,684 negative tweets, and 7,523 neutral tweets. It is important to note that this dataset is restricted to the Egyptian dialect, which is spoken by approximately 100 million Egyptian citizens, and does not encompass any other dialects (Nabil, Aly & Atiya, 2015).

Tweets Emoji Arabic Dataset (TEAD): is collected from X between June 1 and November 30, 2017. It is annotated and compiled using emoji and emotion lexicons, encompassing a total of 5,615,943 tweets classified as positive, negative, or neutral. Among these, 3,122,615 tweets are positive, 2,115,325 are negative, and 378,003 are neutral. The dataset features four distinct Arabic dialects: the Egyptian dialect, Gulfian dialect (which includes six countries: Saudi Arabia, Oman, the United Arab Emirates, Kuwait, Qatar, and Bahrain), Maghrebian dialect (covering five countries: Mauritania, Morocco, Algeria, Tunisia, and Libya), and Levantine dialect (including Syria, Lebanon, Jordan, and Palestine). This dataset is recognized as the most extensive dataset available for Arabic sentiment analysis and can be accessed at https://github.com/HSMAabdellaoui/TEAD?tab=readme-ov-file (Abdellaoui & Zrigui, 2018).

A Twitter-based Benchmark Arabic Sentiment Analysis Dataset (ASAD): comprised 95,000 Arabic tweets, each accompanied by its corresponding sentiment classification. It is categorized into three sentiment groups: Positive, Negative, and Neutral, consisting of 15,215 positive tweets, 15,267 negative tweets, and 64,518 neutral tweets. The tweets are written in various Arabic dialects, including Egyptian, Hijazi, Khaleeji, and Modern Standard Arabic. They are randomly collected using the X API between May 2012 and April 2020 and subsequently annotated by a team of native Arabic speakers. To ensure consistency, each tweet is independently annotated an average of three times. The Arabic Sentiment Analysis Dataset (ASAD) is available at https://www.kaggle.com/code/asalhi/arabic-sentiment-analysis-2nd-place-winning-code/input (Alharbi et al., 2021).

Table 1 Details of the three used datasets.

Dataset name	Dialect	#Positive tweets	#Negative tweets	#Neutral tweets	Source	
ASTD (Nabil, Aly & Atiya, 2015)	Egyptian	799	1,684	7,523	X	
TEAD (Abdellaoui & Zrigui, 2018)	Egyptian	3,122,615	2,115,325	378,003	X	
Maghrebian	
Gulfian	
Levantiane	
ASAD (Alharbi et al., 2021)	Egyptian	15,215	15,267	64,518	X	
Hijazi	
Khaleeji	
Modern Standard Arabic (MSA)	

The statistical analysis presented in Table 1 reveals a notable imbalance within the datasets, which could significantly hinder the performance of binary classification tasks. To address this issue, we established balance by ensuring equal representation of both negative and positive tweets. This balancing approach is executed during the pre-processing stage, focusing solely on positive and negative sentiments while excluding neutral tweets from our analysis. Consequently, all experiments conducted in this study are carried out on the balanced versions of these three datasets.

Tweets pre-processing

The pre-processing of tweets is a crucial phase in conducting sentiment classification, particularly due to the nuances of the Arabic language and its diverse dialects. This stage involves converting unstructured tweet data into a clean, organized format that is compatible with machine learning (ML) and deep learning (DL) methods. To facilitate this, three standard datasets—ASTD, TEAD, and ASAD—are used, focusing solely on positive and negative tweets while excluding neutral ones. To maintain balance, the number of tweets in the positive and negative categories is adjusted to be equal. Essential pre-processing tasks include cleaning the tweets, tokenizing the content, removing stop words, and standardizing the text by taking out diacritics and tatweel. Furthermore, it involves normalizing repeated characters, translating numerical values to Arabic, conducting transliteration, and correlating emojis with appropriate Arabic sentiment expressions. Effectively addressing these pre-processing challenges can substantially improve data quality, reduce noise, and enhance the performance of sentiment analysis systems, resulting in more precise sentiment classification and richer insights into public sentiment across various Arabic dialects on X. The details of the pre-processing phase, including its sub-steps, are outlined in the subsequent sections, with an overview available in Fig. 2. 1. Tweets cleaning: this step requires the deletion of unwanted elements, including links such as http://www.X.com, user mentions, special characters e.g., ($&|−…), punctuation marks e.g. (.: “” ; ’), specific terms such as RT that denotes retweet, hashtags such as #WorldCup2022, as well as any extra spaces that may be present.

2. Tokenization: Tokenization is the process of segmenting a continuous text into smaller components, such as words or phrases. Each of these components is known as a token, which possesses its own distinct meaning and can be employed in later stages of sentiment classification.

3. Stop words elimination: this step entails the elimination of frequently occurring words, such as “فى”, من and “إلي” or “from”, “in”, “to” respectively, which do not carry semantic significance or impact the polarity of the sentence. In contrast, negation terms such as “لا”, “ليس”, and “مش” which translates “not”, “no”, “isn’t” in English are preserved, as they play a crucial role in determining sentence polarity. This methodology facilitates a concentration on more pertinent terms, thereby minimizing the dimensionality of the dataset. To implement this step, a comprehensive list of 1,177 stop words, encompassing various Arabic dialects, is employed.

4. Normalization: is a vital process that converts words into their standardized forms. It typically includes several steps, such as: Removing diacritics: e.g., “الجَمِيلَة” or “aljamila” to “الجميلة or “aljameela””.

Removing tatweel: which means stretching a specific letter within an Arabic word e.g., “بـــــــــــــــــارك” to “بارك”.

Replacing some Arabic characters e.g., (ة ى إأآ ؤ ڳ ٺ) into (ه ي ا و ك ت) to standardize the forms of letters.

Normalizing repeated letters: e.g., “زيااااادة” or “ziyaaada” to “زيادة” or “ziyada”.

Replacing numbers to Arabic words: e.g., “29” to “تسعة وعشرون” or “tisa waashroun”.

5. Transliteration: The goal of this step is to transform all non-Arabic terms into Arabic exclusively. Arabizi is an informal mode of communication used in countries such as Morocco, Tunisia, and Algeria, relying on Latin script. This phenomenon arose due to insufficient technological support for the Arabic language, gaining popularity among Arabic speakers for online interactions (Guellil et al., 2021). In this context, Latin characters or combinations of Latin letters and numbers are used to depict Arabic words. However, the prevalence of Arabizi complicates the comprehension and analysis of Arabic text content. Therefore, it’s essential to convert these terms into their Arabic equivalents. For example, the term “7afla,” which translates to “party” in English should be rendered as “حفلة” or “Hafla”.

6. Mapping emojis into arabic sentiment words: In this step, X users frequently utilize emoticons, known as emojis, to convey their emotions and sentiments, which serve as vital sentiment indicators. This underscores the necessity for an Arabic emoji mapping lexicon that can effectively substitute these visual symbols with appropriate Arabic sentiment words. To accomplish this, a thorough extraction process is performed on the three benchmark datasets employed in the research to identify and classify all positive and negative emojis, while dismissing neutral ones. Sentiment words corresponding to the identified emojis are derived using ChatGPT-4. The resultant lexicon consists of 463 widely used emojis, each accompanied by its relevant description and sentiment term. A visual representation of the developed emoji mapping lexicon is presented in Fig. 3.

Figure 2 Example of tweets pre-processing steps.

Figure 3 Sample emojis from the built emoji mapping lexicon.

Feature engineering

Extracting features is a pivotal phase in both machine learning and deep learning sentiment classification tasks, as it entails gathering valuable information from text data that directly influences the model’s performance. During this phase, word embeddings are employed to transform the words within the pre-processed tweets into vector representations. The feature engineering techniques can be categorized into frequency-based embeddings and prediction-based embeddings (Matrane, Benabbou & Sael, 2023). Frequency-based embeddings depend on the occurrence of each word in the corpus to represent them as vectors. However, despite their ease of computation, these embeddings fail to capture the semantic or grammatical relationships between words. In contrast, prediction-based techniques aim to account for these semantic and syntactic relationships by utilizing neural network-based continuous bag of words (CBOW) or skip-gram (SG) architectures. This work investigates the following word embedding methods for feature extraction: TF-IDF in the context of frequency-based embeddings for machine learning models, and AraVec, FastText, and AraBERT for deep learning models. A detailed discussion of these techniques follows. Term Frequency Inverse Document Frequency (TF-IDF): is a statistical approach designed to assess the relevance of a word within a specific document in relation to a broader set of documents (corpus). It consists of two main parts: Term Frequency (TF), which measures how often a specific term appears in the document, and Inverse Document Frequency (IDF), which indicates the rarity or commonness of that term across the entire corpus (Swamy & Gorabal, 2020). This method is particularly advantageous for word embedding, as it converts text into a numerical format that captures the importance of words, thus improving the efficacy of machine learning models in tasks such as sentiment classification. Notable features of TF-IDF include its ability to emphasize unique words that hold significant meaning in particular contexts, its success in diminishing the influence of common but less informative terms (ex. stop words), and its ease of use. In essence, TF-IDF is a crucial technique for representing text, enabling the extraction of valuable features from unstructured text data for further analysis and categorization tasks.

AraVec: is a sophisticated framework designed specifically for generating word embeddings in the Arabic language, building on the principles of the well-regarded word2vec model. These word embeddings serve as numerical representations that encapsulate the meanings and relationships of words within a continuous vector space, which is vital for various natural language processing (NLP) applications (Masbah, Ababou & Mazroui, 2024). AraVec is distinguished by its extensive training on a vast dataset containing over 3.3 billion tokens from a diverse range of Arabic sources, including tweets, web content, and Wikipedia entries. This comprehensive training enables the model to comprehend the nuanced complexities of the Arabic language, including its dialectical variations. AraVec offers twelve unique configurations for word embeddings, providing flexibility for different use cases. It employs two primary methods: the Skip-gram model, which predicts a target word from its surrounding context, and the Continuous Bag-of-Words (CBOW) model, which uses surrounding words to forecast a target word. This dual approach allows AraVec to effectively recognize and characterize word relationships and similarities, making it an essential technique for the sentiment classification task, where understanding linguistic nuances is critical for precise interpretation. Practically, the two methods are examined with embedding vector size equals 300.

FastText: is an innovative library created by Facebook’s AI Research (FAIR) that enhances the Word2Vec framework by including subword information. This advancement allows FastText to produce word embeddings not just for complete words but also for character n-grams. This feature is particularly beneficial for representing rare words and those not found in the vocabulary, making it suitable for languages with intricate morphological structures (Setyanto et al., 2022). Key features of FastText include its ability to support both supervised and unsupervised learning, which aids in tasks such as text classification and assessing word similarity. The model utilizes hierarchical softmax to improve training efficiency, and its distinct method of representing words as collections of character n-grams helps capture semantic relationships and contextual variances effectively. In our study, we focused on the CBOW variant of FastText, applying position weights, set to a dimension of 300, using character n-grams of length 5, a context window of 5, and including 10 negative samples during the training process.

AraBERT: is a transformer-based model specifically created for understanding the Arabic language, built on the architecture of Bidirectional Encoder Representations from Transformers (BERT). It has undergone pre-training on a large dataset containing 70 million sentences collected from diverse sources, including Wikipedia, Arabic news sites, and extensive collections such as the Abulkhair Arabic Corpus and OSIAN (Antoun, Baly & Hajj, 2020). Several versions of AraBERT exist, and in our study, we focused on BERT-Large-Arabertv2, with a word embedding vector size of 300. This version is notable for its extensive structure, featuring 24 transformer layers, 1,024 hidden units, and 16 attention heads, which allow it to grasp intricate linguistic patterns in Arabic. AraBERT effectively handles both Modern Standard Arabic and various regional dialects. The model utilizes a WordPiece tokenizer tailored to the complex morphological characteristics of the Arabic language, ensuring efficient tokenization.

Review sentiment classification

In this phase, sentiment classification is carried out by utilizing various types of models to achieve optimal results. Specifically, the models include four machine learning-based approaches, two deep learning-based methods, and an ensemble model that combines three transformer-based architectures. These models are finely tuned using the three benchmark dialectal Arabic datasets previously mentioned.

Machine learning-based sentiment classification (ML-based models)

In this step, the models SVM, DT, NB, and XGBoost are assessed to classify dialectal Arabic sentiments into positive and negative labels. The implementation of these models is conducted using the scikit-learn library for machine learning, and a detailed overview of each model follows: Support vector machine (SVM): is a widely-used machine learning technique applicable for both regression and classification problems. It operates by identifying an optimal hyperplane that differentiates various classes within a dataset, aiming to maximize the distance between the classes for better generalization to unseen data. By utilizing a kernel function, SVM can project input data into a higher-dimensional space, allowing for the creation of a linear decision boundary even in cases where the data is not linearly separable (Louati et al., 2023). In this study, SVM is applied with a linear kernel, a regularization parameter of C = 1.0, and a maximum iteration count of 10,000. The gamma parameter is set to ‘auto’ to compute the kernel coefficient automatically.

Decision tree: The DT algorithm is a widely used machine learning method that utilizes a tree-like framework to illustrate features, decisions, and outcomes. This algorithm operates by recursively splitting data based on the most significant features to differentiate classes or estimate the target variable, resulting in a model that is easy to interpret (Alsemaree et al., 2024). In this study, the decision tree classifier is utilized with the default parameters from the scikit-learn library, which includes: criterion = ‘gini’ to indicate a classification task; max_depth = None, allowing the tree to expand until all leaves are either pure or contain fewer than min_samples_split samples; min_samples_split = 2, which is the minimum number of samples needed to split an internal node; and min_samples_leaf = 1, indicating the least number of samples that must be present at a leaf node, which affects the model’s generalization capacity.

Naïve Bayes (NB): is a probabilistic model that employs Bayes’ theorem while assuming that features are mutually independent. Despite this assumption, it has proven effective in various practical applications, including text classification. The classifier estimates the likelihood of a data point belonging to a specific class by calculating the probabilities of its individual features (Suleiman, Odeh & Al-Sayyed, 2022). In this context, the Multinomial Naive Bayes classifier is utilized with its default settings from the scikit-learn library, which are as follows: alpha = 1.0, serving as a smoothing parameter to avoid zero probabilities; fit_prior = True, which indicates that the model will learn the prior probabilities of classes based on the dataset; and class_prior = None, allowing the model to derive the prior probabilities from the data itself.

Extreme gradient boosting (XGBoost): is a powerful machine learning technique that improves predictive modeling via the gradient boosting method. It is designed to efficiently manage large datasets, utilize parallel processing, and incorporate built-in regularization to reduce the risk of overfitting. XGBoost performs exceptionally well with imbalanced datasets and offers the ability to fine-tune hyper-parameters for specific tasks. By sequentially integrating multiple decision trees that correct the mistakes of their predecessors, it forms a robust ensemble model capable of achieving high accuracy in both classification and regression (Afifah, Yulita & Sarathan, 2021). The XGBoost classifier is initially tested with 10 estimators and a learning rate of 1, with the objective function set to ‘binary’, making it suitable for binary classification problems.

Deep learning-based sentiment classification (DL-based models)

CNN and BLSTM models are used in this phase to classify the Arabic opinions. The following subsections explains these models. Convolutional neural network (CNN): represents a formidable category of deep learning architectures primarily designed for processing data with grid-like structures, such as images. They autonomously extract hierarchical features via convolutional layers that capture spatial relationships, complemented by pooling layers that effectively reduce dimensionality while preserving critical information (Ahmad, Saqib & Syed, 2024). The architecture of a CNN is essential, as it significantly influences the model’s learning capability and generalization to new data. Consequently, the selection of layers, activation functions, and hyper-parameters is vital for achieving optimal performance. In this study, as presented in Fig. 4 the CNN architecture is structured with key components starting with an embedding layer that converts input tokens into dense vectors, utilizing a pre-trained embedding matrix loaded with the nn.Embedding.from_pretrained() function. The embedding layer is configured to be non-trainable freeze = True, ensuring the stability of the embeddings during training. The network comprises three convolutional layers featuring kernel sizes of 3, 5, and 7, utilizing the nn.Conv1d function for processing one-dimensional sequences of word embeddings. Each convolutional layer maintains a consistent output channel size of 300, while ReLU activation functions introduce non-linearity to facilitate the capture of complex data patterns. Max pooling operations follow the convolutional layers to extract significant features from each feature map, selecting maxima along the sequence dimension. The outputs from the three convolutional layers are concatenated, yielding a feature vector of size 900. This vector is further processed through two fully connected layers, with the first layer reducing dimensionality from 900 to 150, augmented by a batch normalization layer nn.BatchNorm1d to enhance training stability and generalization. A subsequent ReLU activation introduces additional non-linearity, followed by a dropout layer to mitigate overfitting by randomly nullifying a portion of inputs during training. The final fully connected layer converts the output from 150 to a scalar value indicative of the positive class probability, with a Sigmoid activation applied to constrain the prediction to a probability range of [0, 1]. Thus, the developed CNN model encompasses an embedding layer, three convolutional layers with ReLU activations, max pooling, fully connected layers with batch normalization and ReLU activations, dropout for regularization, and a concluding Sigmoid layer for binary classification.

Bidirectional long short-term memory (BLSTM): The BLSTM architecture represents an advanced recurrent neural network model that integrates LSTM units with bidirectional processing capabilities. This configuration is adept at capturing dependencies in sequential data from both forward and backward perspectives, making it particularly effective for applications such as sequence labeling, speech recognition, and sentiment classification (Elsamadony, Keshk & Abdelatey, 2023). In our investigations, the BLSTM model commenced with an embedding layer derived from a pretrained matrix, which encapsulated word-level semantics. To mitigate overfitting, a dropout layer with a rate of 0.5 is employed. The core of the model consists of a bidirectional LSTM with 128 hidden units, facilitating the acquisition of contextual information from both past and future states within the input sequence. Once the LSTM has processed the data, the model concatenates the final forward and initial backward hidden states, subsequently passing this integrated output through a fully connected layer along with a sigmoid activation function to generate probability scores for binary classification. The implementation of the model is designed to function effectively on both CPU and GPU, allowing for versatile deployment options.

Figure 4 CNN model architecture.

Ensemble learning-based sentiment classification (Transformer-based ensemble model)

Ensemble learning is a powerful technique that combines multiple models, such as classifiers or expert systems, to effectively address particular challenges. This study introduces an ensemble learning approach that merges various transformer models into a unified framework, enhancing the accuracy of sentiment analysis for Arabic text. Our methodology consists of three primary phases: pre-processing, training, and fine-tuning of the foundational models. The following outlines these phases and details the fine-tuning process. Pre-processing: This phase begins with tokenizing the input text into sub-word units utilizing a pre-trained tokenizer. The resulting tokenized output is then transformed into numerical representations, which are suitable for input into the transformer models. Each token is assigned an index from the vocabulary of the pre-trained model, and this index is subsequently converted into tensor format for model consumption.

Baseline models: In the baseline models phase, we concentrate on fine-tuning a selection of pre-trained transformer-based language models, including CAMeLBERT, XLM-RoBERTa, and MARBERT. We remove the default classification layer from these models and replace it with a newly designed layer specific to sentiment classification, distinguishing between positive and negative sentiments. The models are trained using the datasets specified in our research. During this training phase, we utilize a cross-entropy loss function to quantify the discrepancies between predicted and actual sentiment labels, systematically refining the parameters of both the classification layer and the transformer models.

Proposed ensemble approach: Ensemble learning enhances the robustness, generalizability, and predictive accuracy of models across diverse machine learning tasks through various fusion techniques. In our method, we aggregate the outputs from multiple baseline models using an averaging technique, ensuring each fine-tuned model contributes equally. This relationship can be mathematically expressed as follows in Eq. (1):

(1) y^=argmaxi⁡(softmax(1N∑j=1Nwij))

where wj denotes the logit output vector from the j-th classifier, i indicates the index of the i-th class label, and y^ signifies the predicted class label. The aggregated logits are further processed through a softmax layer to generate probability scores. The final sentiment prediction is derived by selecting the class label with the highest score among these aggregated probabilities. This method allows for the effective integration of transformer models while maintaining computational efficiency and adaptability to varied linguistic attributes.

The subsequent section highlights the features of the transformer-based models used in this study. CAMeLBERT: is a language model specifically pre-trained for natural language processing tasks in Arabic. The acronym stands for “Contextualized Arabic Language Model BERT,” with BERT representing “Bidirectional Encoder Representations from Transformers.” CAMeLBERT is designed to address the complexities of the Arabic language, including its diverse dialects and linguistic nuances. The model is available in several versions optimized for Modern Standard Arabic (MSA), Dialectal Arabic (DA), and a combination of both. In our experiments a version called “CAMeL-Lab/bert-base-arabic-camelbert-mix” is used (Inoue et al., 2021). CAMeLBERT has undergone fine-tuning and evaluation across various Arabic NLP tasks, showcasing its effectiveness in processing Arabic text (Inoue et al., 2021).

XLM-RoBERTa: commonly referred to as XLM-R, is a language model built on transformer architecture and pre-trained on text data from 100 languages. It excels in cross-lingual understanding tasks, demonstrating notable improvements over previous multilingual models across various benchmarks. XLM-R effectively manages low-resource languages, achieving significant performance gains in languages such as Swahili and Urdu. It delivers competitive results comparable to robust monolingual models on benchmarks such as GLUE and XNLI, without sacrificing performance in individual languages (Abdul-Mageed, Elmadany & Nagoudi, 2021). Its extensive training on a diverse array of languages, combined with careful optimization of training procedures, significantly enhances cross-lingual representation learning, marking a considerable advancement in multilingual modeling. This study employed a model version called “cardiffnlp/twitter-xlm-roberta-base” (Barbieri, Anke & Camacho-Collados, 2022).

MARBERT: is an advanced transformer-based language model specifically designed for natural language processing tasks in Arabic. Its effectiveness in tackling the challenges posed by the diverse range of Arabic dialects stems from its pre-training on a varied dataset of Arabic tweets. MARBERT’s strong representation of dialectal Arabic is a crucial factor in its success. The model contains approximately 160 million parameters and has been trained for 17 million steps over the course of 36 epochs, making it a highly valuable asset for Arabic natural language processing applications (Conneau et al., 2020). “UBC-NLP/MARBERTv2” model version is utilized in our experiments (Abdul-Mageed, Elmadany & Nagoudi, 2021).

Hyper-parameters optimization techniques

In this phase, two types of hyper-parameter optimization techniques are employed. The first type focuses on hyper-parameter optimization for machine learning models, while the second type addresses hyper-parameter optimization for deep learning and transformer-based models, including their ensemble counterparts. Hyper-parameters optimization techniques for machine learning models: The hyper-parameters for four machine learning models—SVM, DT, NB, and XGBoost—are optimized through Grid Search with five-folds cross-validation, using the Scikit-learn package. Grid search systematically evaluates each parameter combination to identify the optimal settings that enhance model performance, despite the computational expense involved (Shetty et al., 2024). For the SVM, the regularization parameter (C) is tested with values {0.1, 0.5, 0.7, 1, 10, 100, 1,000}, balancing margin maximization and classification error minimization, while the kernel coefficient (gamma) is examined across the values {1, 0.5, 0.4, 0.3, 0.1, 0.01, 0.001, 0.0001, scale, auto} to assess the influence range of individual training samples. The DT model is optimized using three hyper-parameters: max_depth, which determines the maximum depth of each tree, tested with values {None, 10, 20, 30, 40, 50}; min_samples_split, indicating the minimum samples required to split an internal node, assessed with {2, 5, 10}; and min_samples_leaf, representing the minimum samples at a leaf node, evaluated with {1, 2, 4}. For the NB classifier, the regularization strength (alpha) is explored using values {0.1, 0.5, 1.0, 1.5, 2.0}. In XGBoost, various hyper-parameters are fine-tuned, including learning_rate, which regulates the contribution of each tree; n_estimators, specifying the number of boosting rounds; max_depth; min_child_weight, indicating the minimum sum of instance weight necessary in a child node; subsample, denoting the fraction of samples for growing each tree; and colsample_bytree, which specifies the fraction of features used at each tree construction. These hyper-parameters are tested using the following respective values: {0.8, 1.0, 1.2, 1.3, 1.4}, {1, 2, 3, 4, 5}, {8, 10, 12, 13, 14, 15, 16}, {1}, {0.8, 0.9, 1.0, 1.5, 2.0}, and {0.8, 0.9, 1.0, 1.5, 2.0}.

Hyper-parameters optimization techniques for deep learning models and transformer- based models: The hyper-parameters dropout_rate, learning_rate, and weight_decay are optimized to enhance the performance of CNN, BLSTM, and the proposed ensemble model using the Optuna framework. Optuna is a robust library that automates the tuning of hyper-parameters for AI models using the Adam optimizer, featuring flexible designs for efficient optimization, lightweight in-memory operations, and the ability to scale with relational databases. Notable functionalities include exporting results to pandas and offering a web dashboard for real-time data visualization (Akiba et al., 2019). The dropout_rate, which helps mitigate overfitting by reducing dependence on specific neurons, is tested across a range from [0.0, 0.5] in increments of 0.05, with the optimal value determined to be 0.5. The learning_rate is explored on a logarithmic scale between [ 10e−5, 10e−1], ultimately identifying the best rate as 3.3e−3. Additionally, weight_decay is also evaluated on a logarithmic scale within the range [ 10e−4, 10e−1], resulting in an optimal value of 1e−2.

Table 2 summarizes various hyper-parameter optimization techniques used in both machine learning and deep learning models. It outlines the specific hyper-parameters for each model along with the corresponding optimization values. This concise overview underscores the variety of approaches that are crucial for improving model performance and ensuring effective application across diverse datasets and tasks.

Table 2 Summary of hyper-parameter optimization techniques.

Model category	Model	Hyper-parameters	Optimization values	
Machine learning	Support vector machine (SVM)	Regularization parameter (C)	{0.1,0.5,0.7,1,10,100,1,000}	
Kernel coefficient (Gamma)	{1, 0.5, 0.4, 0.3, 0.1, 0.01, 0.001, 0.0001, scale, auto}	
Decision tree (DT)	Max depth	{None, 10, 20, 30, 40, 50}	
Min samples split	{2, 5, 10}	
Min samples leaf	{1, 2, 4}	
Naive Bayes (NB)	Regularization strength (Alpha)	{0.1, 0.5, 1.0, 1.5, 2.0}	
XGBoost	Learning rate	{0.8, 1.0, 1.2, 1.3, 1.4}	
Number of estimators ( nestimators)	{1, 2, 3, 4, 5}	
Max depth	{8, 10, 12, 13, 14, 15, 16}	
Min child weight	{1}	
Subsample	{0.8, 0.9, 1.0, 1.5, 2.0}	
Colsample by tree	{0.8, 0.9, 1.0, 1.5, 2.0}	
Deep learning and transformer-based learning	CNN, BLSTM, and ensemble model	Dropout rate	Range: [0.0, 0.5] (increments of 0.05)	
Learning rate	Logarithmic scale: [10e−5, 10e−1]	
Weight decay	Logarithmic scale: [10e−4, 10e−1]	

The analysis of data presented in Table 2 highlights the critical role of hyper-parameter optimization in model training, directly affecting performance outcomes. Models like support vector machines and decision trees depend on specific parameters to enhance generalization, while deep learning architectures such as CNN and BLSTM require careful tuning of learning rates and dropout rates to mitigate overfitting. The diversity of optimization techniques emphasizes the need for tailored strategies for different model types, underscoring that a uniform approach is inadequate in both machine learning and deep learning applications. Effective hyper-parameter tuning ultimately results in more robust algorithms that yield superior performance in real-world tasks.

Results and analysis

This section provides a detailed discussion of the conducted experiments, including their results and the evaluation metrics employed.

Performance metrics

To evaluate the performance of the models employed for the dialectal Arabic sentiment classification task, four standard performance metrics are utilized: accuracy, precision, recall, and F1-score. Accuracy (Acc) Eq. (2) is determined by dividing the number of correctly classified predictions by the total number of predictions. The precision (Pr) metric Eq. (3) is calculated by dividing the number of correctly classified positive instances by the total number of positively classified instances. Recall (R) Eq. (4) is computed by dividing the number of correctly classified positive instances by the total number of instances classified. Lastly, the F1-score Eq. (5) is a performance metric that incorporates both precision and recall, providing a balanced measure of the model’s accuracy.

(2) Acc=TP+TNTP+TN+FP+FN

(3) Pr=TPTP+FP

(4) R=TPTP+FN

(5) F1−score=2×recall×precisionrecall+precision

The formulas mentioned above utilize the following terms: true positive (TP) refers to the number of accurately predicted positive values, while false positive (FP) indicates the count of instances incorrectly classified as positive despite being negative. False negative (FN) represents the number of instances inaccurately classified as negative when they are, in fact, positive. Lastly, true negative (TN) signifies the count of accurately predicted negative values.

Experimental results

The proposed sentiment classification process for Dialectal Arabic is assessed through three distinct types of experiments—machine learning-based, deep learning-based, and transformer-based ensemble experiments—conducted before and after the hyper-parameter optimization process. These experiments utilize three previously mentioned benchmark datasets, namely ASTD, ASAD, and TEAD. To evaluate both machine learning and deep learning approaches, various feature extraction techniques are employed, including AraVec, FastText, AraBERT, and TF-IDF. In the machine learning experiments, shallow models such as NB, SVM, XGBoost, and DT are utilized alongside the TF-IDF feature extraction method, with their performance scores presented in Table 3.

Table 3 Performance of machine learning models using the three datasets before hyper-parameters optimization.

Model	Dataset	Accuracy (%)	Precision (%)	Recall (%)	F1-score (%)	
SVM	TEAD	77	76	77	76.5	
ASAD	92	92	92	92	
ASTD	74	73	74	73.5	
Average	81	80.3	81	80.7	
NB	TEAD	73	74	73	73.5	
ASAD	88	88	88	88	
ASTD	72	75	72	73.5	
Average	77.7	79	77.7	78.3	
DT	TEAD	72	72	72	72	
ASAD	87	87	87	87	
ASTD	70	69	70	69.5	
Average	76.3	76	76.3	76.2	
XGBoost	TEAD	71.7	70	71	70.5	
ASAD	84	87	84	85.5	
ASTD	71	70	71	70.5	
Average	75.6	75.7	75.3	75.5	

As illustrated in Table 3, the SVM model outperformed the other models across the three benchmark datasets, achieving average results of 81%, 80.3%, 81%, and 80.7% for accuracy, precision, recall, and F1-score, respectively.

To enhance performance, the machine learning models are optimized using grid search, and the results are presented in Table 4.

Table 4 Performance of machine learning models using the three datasets after hyper-parameters optimization.

Model	Dataset	Accuracy (%)	Precision (%)	Recall (%)	F1-score (%)	
SVM	TEAD	92	92	92	92	
ASAD	92	92	92	92	
ASTD	71	73	71	72	
Average	85	85.7	85	85.3	
NB	TEAD	74	73	74	73.5	
ASAD	89	89	89	89	
ASTD	70	73	70	71.5	
Average	77.7	78.3	77.7	78	
DT	TEAD	70.6	63	50	55.8	
ASAD	92	92	92	92	
ASTD	69	69	69	69	
Average	77.2	74.7	70.3	72.3	
XGBoost	TEAD	70.7	72	50	59	
ASAD	87	87	87	87	
ASTD	71	70	71	70.5	
Average	76.2	76.3	69.3	72.2	

Table 4 demonstrates that after the hyper-parameter optimization process, the SVM model consistently outperformed all other models, showcasing an increase of 4% in average accuracy, 5.4% in average precision, 4% in average recall, and 4.6% in the average F1-score.

In the deep learning-based experiments, both CNN and BLSTM models are tested using AraVec, FastText, and AraBERT feature extraction methods. The results for the CNN model are presented in Table 5.

Table 5 CNN performance results using the three datasets before hyper-parameters optimization.

Model	Dataset	Accuracy (%)	Precision (%)	Recall (%)	F1-score (%)	
AraVec-based (COW)	TEAD	82.7	81	74	77.3	
ASAD	92.2	92	92	92	
ASTD	81	78	77	77.5	
Average	85.3	83.7	81	82.3	
AraVec-based (SG)	TEAD	82.4	79	76	77.5	
ASAD	93	93	93	93	
ASTD	83.2	80	80	80	
Average	86.2	84	83	83.5	
FastText	TEAD	83.5	83	75	78.8	
ASAD	92.2	92	92	92	
ASTD	81.3	78	77	77.5	
Average	85.7	84.3	81.3	82.8	
AraBERT	TEAD	80	76	77	76.5	
ASAD	93	93	93	93	
ASTD	80.7	77	79	78	
Average	84.7	82	83	82.5	

As shown in Table 5, the AraVec-based skip-gram (SG) method outperformed the other feature extraction techniques, achieving the best performance with the CNN model. It recorded an average accuracy of 86.2%, an average precision of 84%, an average recall of 83%, and an average F1-score of 83.5% when applied to the three datasets.

Building on the results presented in Table 5 for the CNN model utilizing the AraVec-based skip-gram (SG) feature extraction method, the Adam optimizer is employed to enhance performance. The outcomes of this optimization are detailed in Table 6.

Table 6 Results of CNN with AraVec-based skip gram after optimization using the three datasets.

Dataset	Accuracy (%)	Precision (%)	Recall (%)	F1-score (%)	
TEAD	83.5	83	79	81	
ASAD	93	93	93	93	
ASTD	83.6	81	80	80.5	
Average	86.7	85.7	84	84.8	

As shown in Table 6, the results for the CNN model improved following the optimization process, yielding an increase of 0.5% in average accuracy, 1.7% in average precision, 1% in average recall, and 1.3% in average F1-score.

The BLSTM model is also evaluated using the AraVec, FastText, and AraBERT feature extraction methods, and the findings are presented in Table 7.

Table 7 BLSTM performance results using the three datasets before hyper-parameters optimization.

Model	Dataset	Accuracy (%)	Precision (%)	Recall (%)	F1-score (%)	
AraVec-based (COW)	TEAD	83.7	82	76	79	
ASTD	89.2	89	89	89	
ATSAD	82	80	77	78.5	
Average	85	83.7	80.7	82.2	
AraVec-based (SG)	TEAD	83.3	82	76	79	
ASAD	92.7	92	92	92	
ASTD	82.4	80	77	78.5	
Average	86.1	84.7	81.7	83.2	
FastText	TEAD	85	86	75	78	
ASAD	93	92	92	92	
ASTD	83	80	78	79	
Average	87	86	81.7	83	
AraBERT	TEAD	84.2	86	74	79.6	
ASAD	92.7	93	92	92.5	
ASTD	83	80	80	80	
Average	86.6	86.3	82	84	

The findings presented in Table 7 indicate that the BLSTM model achieved superior performance with the FastText feature extraction method, reaching an average accuracy of 87%, an average precision of 86%, an average recall of 81.7%, and an average F1-score of 83%.

The BLSTM model, utilizing the FastText feature extraction method, is optimized with the Adam optimizer to achieve enhanced performance, with the results presented in Table 8.

Table 8 Results of BLSTM with FastText after optimization using the three datasets.

Dataset	Accuracy (%)	Precision (%)	Recall (%)	F1-score (%)	
TEAD	86.5	86	76	80.7	
ASAD	93.8	93	93	93	
ASTD	84	81	80	80.5	
Average	88.1	86.7	83	84.7	

The BLSTM model utilizing FastText demonstrates improvements in various metrics, including a 1.1% increase in average accuracy, a 0.7% increase in average precision, a 1.3% increase in average recall, and a 1% increase in average F1-score, as indicated in Table 8.

Recently, driven by the widespread popularity of transformer models, several have been tested on the three utilized datasets, specifically CAMeLBERT, XLM-RoBERTa, MARBERT, and their ensemble model. The results of these experiments are presented in Table 9.

Table 9 Performance results of CAMeLBERT, XLM-RoBERTa, MARBERT, and their ensemble model using the three used datasets before hyper-parameters optimization.

Model	Dataset	Accuracy (%)	Precision (%)	Recall (%)	F1-score (%)	
CAMeLBERT	TEAD	85	79	80	80	
ASAD	95	95	95	95	
ASTD	83.2	83	80	81	
Average	87.7	85.7	85	85.3	
MARBERT	TEAD	84	79	81	80	
ASAD	95.3	95	95	95	
ASTD	86.6	86	84	85	
Average	88.6	86.7	86.7	86.7	
XLM-RoBERTa	TEAD	85	80	82	81	
ASAD	94.2	94	94	94	
ASTD	80.7	80	77	78	
Average	86.6	84.7	84.3	84.3	
Proposed ensemble model	TEAD	85.4	80	83	81	
ASAD	95.6	95	95	95	
ASTD	86.6	86	84	85	
Average	89.2	87	87.3	87	

In Table 9, the monolingual model MARBERT exhibited superior performance compared to the standalone models CAMeLBERT and XLM-RoBERTa, achieving an average accuracy of 88.6%, an average precision of 86.7%, an average recall of 86.7%, and an average F1-score of 86.7%. However, the proposed ensemble model, which integrates these three transformer models, delivered even more impressive results, with an average accuracy of 89.2%, an average precision of 87%, an average recall of 87.3%, and an average F1-score of 87%.

To enhance performance, the three transformer models, along with their combined ensemble model, are optimized using the Adam optimizer, and the results are detailed in Table 10.

Table 10 Performance results of CAMeLBERT, XLM-RoBERTa, MARBERT, and their ensemble model using the three used datasets after hyper-parameters optimization.

Model	Dataset	Accuracy (%)	Precision (%)	Recall (%)	F1-score (%)	
CAMeLBERT	TEAD	86.5	80	80	80	
ASAD	95	95	95	95	
ASTD	84.2	83	80	81.5	
Average	88.6	86	85	85.5	
MARBERT	TEAD	85	81	81	81	
ASAD	95.3	95	95	95	
ASTD	87.6	86	84	85	
Average	89. 3	87.3	86.7	87	
XLM-RoBERTa	TEAD	85.7	80	82	81	
ASAD	94.2	94	94	94	
ASTD	83.7	80	77	78.5	
Average	88	84.7	84.3	84.5	
Proposed ensemble model	TEAD	87.4	83	83	83	
ASAD	95.6	95	95	95	
ASTD	88.3	86	84	85	
Average	90.4	88	87.3	87.7	

As shown in Table 10, after optimization, the proposed ensemble model continued to lead, outperforming all other models used in the conducted experiments. It achieved an average accuracy of 90.4%, an average precision of 88%, an average recall of 87.3%, and an average F1-score of 87.7%.

A performance comparison of the evaluation models employed in this study, both before and after hyper-parameter optimization, is illustrated in Figs. 5 and 6, respectively.

Figure 5 Performance comparison between models before hyper-parameters optimization.

Figure 6 Performance comparison between models after hyper-parameters optimization.

The proposed transformer-based ensemble model, as shown in both Figs. 5 and 6, exhibits outstanding performance in sentiment classification of dialectal Arabic tweets by leveraging the combined strengths of multiple models to enhance overall effectiveness.

Table 11 and Fig. 7 provide a comparison of the performance between the proposed transformer-based ensemble model and the preceding methods for each dataset used.

Table 11 Performance comparison between the proposed model and the prior studies.

Study	Dataset	Dataset status	F1-score (%)	
Abdellaoui & Zrigui (2018)	TEAD	Unbalanced	81.9	
Mohamed et al. (2022)	ASAD	79.986	
El Karfi & El Fkihi (2022)	ASTD	84.48	
The proposed transformer-based ensemble model	TEAD	Balanced	83	
ASAD	95	
ASTD	85	

Figure 7 Performance comparison between the proposed model and the prior studies (Mohamed et al., 2022; El Karfi & El Fkihi, 2022; Abdellaoui & Zrigui, 2018).

The findings displayed in Table 11 and Fig. 7 indicate that the proposed transformer-based ensemble model, which is trained on a balanced version of each dataset, exceeds the performance of earlier models that were trained on unbalanced datasets regarding the F1-score. This improvement is attributed to the fact that class imbalance usually results in a preference for the majority class by the models, which consequently diminishes the F1-score for the minority class. By utilizing a balanced dataset for training, the proposed model is able to learn more effectively from both classes, minimizing bias and enhancing the F1-score, subsequently leading to superior overall performance.

Results discussion

The experiments conducted to classify sentiment in Dialectal Arabic demonstrated a significant enhancement in model performance across three distinct stages: traditional machine learning, deep learning, and ensemble techniques utilizing transformer architectures. These studies are carried out using balanced versions of three benchmark datasets—TEAD, ASTD, and ASAD—with a focus on binary classification to ensure a comprehensive representation of both negative and positive sentiment, thereby strengthening the reliability of the evaluation process. The initial phase involved examining several machine learning algorithms, including SVM, DT, NB, and XGBoost, employing TF-IDF for feature extraction, with SVM proving to be the most effective model. It achieved an average accuracy of 81%, along with precision, recall, and F1-scores of 80.3%, 81%, and 80.7%, respectively. After implementing hyper-parameter optimization through grid search, SVM showed significant improvements, realizing increases of 4% in accuracy, 5.4% in precision, 4% in recall, and 4.6% in F1-score. In the subsequent phase, the evaluation of deep learning models, particularly CNN and BLSTM, is conducted using feature extraction techniques such as AraVec, FastText, and AraBERT. The CNN model achieved an accuracy of 86.2%, with corresponding precision, recall, and F1-scores of 84%, 83%, and 83.5%. Post-optimization, the CNN experienced modest improvements, with a 0.5% increase in accuracy and enhancements in precision, recall, and F1-score. In contrast, the BLSTM model excelled with FastText, attaining an average accuracy of 87%, with precision, recall, and F1-scores of 86%, 81.7%, and 83%, respectively. After optimization, BLSTM recorded notable advancements, particularly a 1.1% increase in accuracy. The highest level of performance is reached with the transformer-based ensemble model, which integrated CAMeLBERT, XLM-RoBERTa, and MARBERT, resulting in a considerable boost in classification performance and achieving an average accuracy of 90.4%, alongside precision, recall, and F1-scores of 88%, 87.3%, and 87.7%, respectively. The proposed transformer-based ensemble model is systematically compared to alternative methodologies using the three specified datasets. For the TEAD dataset, the ensemble model demonstrated superior performance over the SVM model outlined in Abdellaoui & Zrigui (2018), achieving notable performance metrics of 87.4% accuracy, 83% precision, 83% recall, and an F1-score of 83%. In contrast, the SVM model, which utilized TF-IDF as a feature extractor, recorded an F1-score of 81.9% as mentioned in Abdellaoui & Zrigui (2018). When evaluating the ASAD dataset, the ensemble model reached peak performance with an accuracy of 95.6%, precision of 95%, recall of 95%, and an F1-score of 95%, markedly surpassing the F1-score of 79.986% reported in Mohamed et al. (2022). Similarly, for the ASTD dataset, the proposed model outperformed the one presented in El Karfi & El Fkihi (2022), achieving an accuracy of 88.3%, precision of 86%, recall of 84%, and an F1-score of 85%, compared to the accuracy of 86.52% and F1-score of 84.48% of the referenced model. The results presented in our study underscore the advantages of utilizing a balanced dataset, which enhances the model’s ability to learn effectively from both positive and negative sentiment classes, and the efficacy of ensemble methods in harnessing the strengths of multiple advanced models. This reinforces the capabilities of transformer architectures in addressing the complexities inherent in sentiment analysis of Arabic dialects.

Conclusion and future work

In conclusion, this study provides an in-depth analysis of progress made in sentiment classification within dialectal Arabic, showcased through a series of thorough experiments that incorporate machine learning, deep learning, and transformer-based ensemble approaches. The investigation assessed various machine learning models, including SVM, NB, DT, and XGBoost, with the SVM standing out as the most effective, achieving an average accuracy of 81% before any optimization. In the deep learning segment, both CNN and BLSTM models are utilized, yielding notable results: the CNN with AraVec extracted features achieved an average accuracy of 86.2%, while the BLSTM using FastText reached 87%. Subsequently, the implementation of an ensemble model that combined the strengths of CAMeLBERT, XLM-RoBERTa, and MARBERT resulted in remarkable performance, with the optimized ensemble attaining an impressive average accuracy of 90.4%. Moreover, the research contributed to the field by creating an Arabic emoji mapping lexicon for improved feature engineering and utilized the Optuna library with the Adam optimizer for hyper-parameter optimization, establishing a new standard for sentiment classification in dialectal Arabic. These results not only pave the way for future research but also provide valuable insights into the efficacy of various modeling techniques for analyzing sentiments in the Arabic language.

Our future strategy involves training a large language model for aspect-based sentiment classification task in dialectal Arabic.

Supplemental Information

Supplemental Information 1 Code.

Supplemental Information 2 Raw data.

Supplemental Information 3 Computing infrastructure specification.

Supplemental Information 4 Full guide for coding scripts.

Additional Information and Declarations

Competing Interests

Omar Mansour, Eman Aboelela, Remon Talaat, and Mahmoud Bustami are employed by Intella.

Author Contributions

Omar Mansour conceived and designed the experiments, authored or reviewed drafts of the article, and approved the final draft.

Eman Aboelela conceived and designed the experiments, performed the experiments, analyzed the data, performed the computation work, prepared figures and/or tables, authored or reviewed drafts of the article, and approved the final draft.

Remon Talaat performed the experiments, analyzed the data, performed the computation work, prepared figures and/or tables, and approved the final draft.

Mahmoud Bustami performed the experiments, performed the computation work, prepared figures and/or tables, and approved the final draft.

Data Availability

The following information was supplied regarding data availability:

The raw data and code are available in the Supplemental Files.

The TEAD: Large Scale Arabic Dataset for Sentiment Analysis is available at GitHub: https://github.com/HSMAabdellaoui/TEAD?tab=readme-ov-file.

The ASTD is available at GitHub: https://github.com/mahmoudnabil/ASTD/tree/master/data.

The Arabic Sentiment Analysis is available at Kaggle: https://www.kaggle.com/code/asalhi/arabic-sentiment-analysis-2nd-place-winning-code/input.

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
