# Peer review of "Transformer-based ensemble model for dialectal Arabic sentiment classification"

_PeerJ Computer Science, doi:10.7717/peerj-cs.2644_

## Round 0.1 · original submission · Major Revisions

Summary of Decision: Based on the reviewers' feedback, I recommend that the manuscript undergo major revisions before reconsideration for publication. The paper demonstrates interesting contributions to sentiment analysis and ensemble learning approaches. However, significant issues concerning clarity, detailed methodology, and handling of data imbalance must be addressed.

Key Reasons for Major Revisions:
1. Basic Reporting:
o Language and Academic Style: Reviewer 1 pointed out that the manuscript's English needs improvement, including grammar and consistency in tense usage. Improving clarity through editing for grammar and academic tone is essential.
o Terminology Consistency: Reviewer 3 mentioned inconsistencies in terminology, such as "Tweeter" versus "X." Consistent terminology and correct usage are critical to maintaining professional standards in academic writing.
2. Experimental Design:
o Details of Methods: Reviewer 1 highlighted that several aspects of the experimental design lack crucial detail. Specifically:
 There is insufficient explanation of the word embedding techniques (e.g., AraVec, FastText, AraBERT). The authors need to elaborate on these embeddings, their characteristics, and why each one was chosen.
 The preprocessing steps, especially regarding the handling of stopwords, should be clarified, as this can greatly impact the sentiment classification outcomes.
 Important details regarding the hyperparameter tuning process and the architecture of models like CNN must be included. The current level of detail is insufficient for reproducibility.
 The ensemble method is not well-described. Reviewer 1 stressed the need for a clear explanation of how the models were combined (parallel or cascade), the weights assigned, and the methodology used for blending results. This is essential for understanding the robustness of the ensemble approach.
3. Validity of Findings:
o Data Imbalance: Reviewer 1 noted a significant data imbalance in the TEAD dataset, which could impact the validity of the model's performance. The authors should address this issue by providing details about any balancing techniques applied or by discussing the limitations of this imbalance.
o Comparison with Previous Works: The manuscript lacks a section that compares the findings to previous studies. Including a discussion that highlights how the proposed methods compare with state-of-the-art approaches would strengthen the contribution and situate the work more effectively within the existing literature.
4. Reviewer Feedback and Recommendations:
o Reviewer 1 (Major Revisions): Provided detailed feedback on issues with basic reporting, experimental design, and the need for methodological clarification. Addressing these issues is crucial.
o Reviewer 2 (Accept): Recommended acceptance without major concerns, noting the manuscript's clear presentation and sound experimental setup. However, given the detailed feedback from the other reviewers, further improvements are still required.
o Reviewer 3 (Minor Revisions): Suggested minor revisions, including updating references to be more recent and ensuring consistent terminology throughout the paper. The discussion on recent advances and future directions is appreciated and should be retained while addressing these smaller points.
Recommendations for Authors:
1. Language Editing: Revise the manuscript for grammatical accuracy and consistency in academic tone. Consider professional language editing.
2. Detailed Methodology: Expand the descriptions of:
o Word embedding techniques and hyperparameter tuning.
o The ensemble learning strategy—clarify how models were combined, their respective weights, and the blending process.
o Data preprocessing, particularly concerning stopwords, and justify the choices made.
3. Data Imbalance: Address the dataset imbalance by either applying rebalancing techniques or discussing its impact on model performance and potential mitigation strategies.
4. Comparison with Previous Works: Add a section to compare the proposed methods to similar existing works to establish the novelty and contribution of this work.
5. Terminology Consistency and Updated References: Ensure consistency in terminology (e.g., replace "Tweeter" with "X") and update older references as suggested by Reviewer 3.

The manuscript's contributions to sentiment analysis and ensemble learning approaches are promising. However, addressing the above points is crucial for enhancing the quality, reproducibility, and clarity of the work. I look forward to reviewing a revised version that incorporates these substantial improvements.

Reviewer 1 ·

Basic reporting

- English must be improved. Check and improve the grammar of the entire text.
- In academic writing, use the present tense and the third person. Avoid the past and future tense.
- In academic writing: Do not use , “so”. use “therefore/hence/thus...”

Experimental design

1- The Data Preprocessing step lacks sufficient explanation. It is essential to provide details about the employed word embedding techniques (AraVec, FastText, AraBERT), including their characteristics.

2- In the analysis of dialects, and more specifically for sentiment classification, several studies do not recommend eliminating stopwords. These words play a crucial role in the meaning of texts, and sometimes the presence or absence of a one word can reverse the sentiment polarity from positive to negative or vice versa.

3- How did you tune the hyper-parameters? Add a summary of the values of these hyperparameters.

4- The implementation of the ML algorithms is unclear, and their parameters used were not provided.

5- Furthermore, the explanations of the other models presented are lacking in detail. e.g:
• The CNN model is only presented as a definition, and it is cited with an incorrect reference (Convolutional Neural Network (CNN) (Priyadarshini and Cotton, 2021)). The authors must provide a comprehensive description of the model, including its architecture. This should encompass the different layers, inputs, outputs, and activation functions used. This detailed presentation is essential for a clear understanding of the model's structure and functionality.

• In the section "Ensemble Learning-Based Sentiment Classification," the authors mention, "We employed a three-fold ensemble approach using fine-tuned transformer-based models, namely CAMeLBERT, XLM-RoBERTa, and MARBERT. We selected the predictions with the highest scores from each model and applied this process to all datasets. Finally, we used a Blending Ensemble technique to integrate the top-performing submissions, resulting in our high final score." However, the description lacks important details. The authors must elaborate on how the models were assembled. Specifically, they should clarify:
- How the models were combined: Was it in parallel or in a cascade?
- The weights assigned to each model in the ensemble.
- The type of output vectors used and how they were processed.
- The specific methodology and criteria for selecting the highest scores and the blending process.
Providing these details is crucial for understanding the effectiveness and reproducibility of the ensemble approach.

Validity of the findings

1- The dataset is not balanced, particularly in the case of TEAD, where the number of positive tweets is 3,122,615 and the number of negative tweets is 2,115,325, resulting in a difference of 1,007,290 tweets (47.61%). This imbalance significantly impacts the classification results, especially in binary classification tasks.

2- The conclusion are not well conducted, it should use brief words to enhance and highlight the contribution and novelty.

3- Missing a section to present and discuss the comparison with previous works.

Additional comments

Please include brief documentation on how to use the code for replication and research purposes.

Cite this review as

Reviewer 2 ·

Basic reporting

no comment

Experimental design

The authors made very good experimental and wrote very well.

Validity of the findings

'no comment

Additional comments

I accept to publish this paper

Cite this review as

Reviewer 3 ·

Basic reporting

Overall, the paper makes a significant contribution to the field of sentiment analysis by providing a comprehensive Depth of Analysis. It is well-organized and clearly written, with a thorough analysis of recent advances and future directions. I recommend accepting the paper with minor revisions to address the suggestions for improvement.

Experimental design

Gaps in Review: 1. Tweeter may be replaced with “X”. 2. Use “Figure/Fig. ” suggested for using any one in the article. 3. Automatic opinion mining and sentiment analysis are two different aspects. So clear discussions are essentials for both the context.

Validity of the findings

Clarity and Detail: Some sections, such as the taxonomy, could be enhanced with more detailed explanations and examples.
1. Tweets Pre-processing may be discussed with own angle. Eliminating undesired elements may lead to rise different meanings and may cause separation in sentiments.
2. Suggested for citation in Figure 3.
3. Older references, before 2018 may be discarded, as it may faint the accuracy. New references are suggested for citation.

Additional comments

• Recent Advances: The discussion on recent advances is up-to-date and integrates well with the overall classification together with the Dialectal Sentiment Classification .
• Future Directions: The proposed future research directions are valuable and can profit subsequent social media platforms.

Cite this review as

---

## Round 0.2 · Minor Revisions

Pay attention to the reviewers' comments.

Reviewer 1 ·

Basic reporting

Thank you for addressing some of the points raised during the first round of review. However, there are still critical issues that need to be addressed before this manuscript can be considered for publication. Please see the detailed comments below:

Experimental design

Thank you for addressing some of the points raised during the first round of review. However, there are still critical issues that need to be addressed before this manuscript can be considered for publication. Please see the detailed comments below:

1. Hyper-parameters Optimization Techniques:
Please include a table summarizing the Hyper-parameters Optimization Techniques for Machine Learning, Deep Learning, and Transformer-based models (Page 15). Providing this will help improve the clarity and comprehensiveness of the manuscript.

2. CNN Model Architecture:
For better readability and understanding, I recommend presenting the CNN model architecture, including its various layers, using a flowchart. This will enhance the visualization of your model and make it easier for readers to grasp its structure.


3. Ensemble Learning-Based Sentiment Classification:
In the authors' response, they stated: "More details are added to the section of 'Ensemble Learning-Based Sentiment Classification (Transformer-based Ensemble Model)' on pages 14-15." However, upon reviewing this subsection, no additional details were found. The text still only describes the three transformer models and their characteristics, without providing specifics on how they were combined, including weight assignments, output vector processing, or the methodology used for selecting the highest scores and blending. Please address these missing details to enhance clarity.

Validity of the findings

4. Comparison with Previous Works:
The following previous review point has not yet been addressed: "Missing a section to present and discuss the comparison with previous works."
+ It is essential to include a dedicated subsection comparing your results with prior studies. This will provide context for your findings and demonstrate the novelty and effectiveness of your approach.

Addressing the above points will significantly strengthen the manuscript and ensure it meets the journal's standards.

Annotated reviews are not available for download in order to protect the identity of reviewers who chose to remain anonymous.
Cite this review as

---

## Round 0.3 · accepted · Accept

The authors have addressed all of the reviewer's comments.
Congratulations.

Reviewer 1 ·

Basic reporting

no comment

Experimental design

no comment

Validity of the findings

no comment

Cite this review as